The association between lifetime cigarette smoking and dysphonia in the Korean general population: findings from a national survey

Byeon Haewon byeon@nambu.ac.kr
Department of Speech Language Pathology & Audiology, Nambu University , Gwangju , Republic of Korea
Speech-Language Pathology Center, Nambu University , Gwangju , Republic of Korea
Lo Presti Alessandra
Electronic publication date: 2015 Apr 28
Publication date: 2015
Volume: 3
Electronic Location ID: e912
Received 2015 Feb 8; Accepted 2015 Apr 5
Copyright: © 2015 Byeon
Copyright year: 2015
Copyright holder: Byeon
License: This is an open access article distributed under the terms of the Creative Commons Attribution License, which permits unrestricted use, distribution, reproduction and adaptation in any medium and for any purpose provided that it is properly attributed. For attribution, the original author(s), title, publication source (PeerJ) and either DOI or URL of the article must be cited.
License URL: https://creativecommons.org/licenses/by/4.0/

Keywords: Voice problems, Pack years, Cigarette smoking, Dysphonia, Laryngeal disorders

Funding: The author declares there was no funding for this work.

==============================
This study aims to investigate the relationship between current smoking and lifetime amount smoked and the incidence of dysphonia using data from a national cross-sectional survey that represents the Korean population. Subjects were 3,600 non-institutionalised civilian adults over the age of 19 (1,501 males and 2,099 females) who completed the laryngeal examination of the 2008 Korea National Health and Nutrition Examination Survey (KNHANES). For lifetime amount smoked, subjects were classified as light smokers (≤26.7 pack years), medium smokers (26.7–40.5 pack years), heavy smokers (40.5–55.5 pack years), and very heavy smokers (55.5–156 pack years) based on pack years (packs smoked per day × years as a smoker). The odds ratio (OR) for the statistical test was presented using hierarchical logistic regression. When adjusted for covariates (age, gender, level of education, income, occupation, alcohol consumption and pain/discomfort during the last two weeks), current smokers had a 1.8 times (OR = 1.77, 95% CI [1.17–2.68]) higher risk for self-reported voice problems than non-smokers. Moreover, current smokers had a 1.6 times (OR = 1.56, 95% CI [1.02–2.38]) higher risk of laryngeal disorder (p < 0.05). In terms of pack years, very heavy smokers were significantly more likely to have laryngeal disorder, while self-reported voice problems were significantly more likely for heavy smokers but not very heavy smokers. The results of this study imply that chronic smoking has a significant relationship with dysphonia. Longitudinal studies are required in future work to verify the causality between lifetime smoking amount and dysphonia.

Introduction

Although the smoking rate of adults in Korea has been steadily decreasing for the last decade, the smoking rate of adult males in Korea as of 2008 is 46.1%, which is the highest among OECD countries (Ministry of Health and Welfare, 2010). Smoking is a major risk factor that exerts a detrimental effect on vocal health. Repeated smoking can lead to the inability to move the vocal cords completely during vocalisation through its harmful impact on the mucosal membrane of the oral cavity, resulting in a deterioration of vocal quality and pitch (Menvielle et al., 2004). An acoustic analysis of the voices of smokers revealed that smokers have a lower fundamental frequency than non-smokers (Awan, 2011; Wiskirska-Woźnica et al., 2004) and their vocal quality is unstable (Vincent & Gilbert, 2012). Moreover, it has been shown that it takes smokers less time before the onset of vocal fatigue than non-smokers and smokers experience more severe vocal fatigue (Park, 2003). In addition, smoking is known as a risk factor for laryngeal disorders. Chronic smoking is a major cause of laryngeal and oral cancer, and it is a risk factor for laryngeal keratosis, Reinke’s edema and laryngeal leukoplakia (Feierabend & Shahram, 2009; Gnjatic, Stankovic & Djukić, 2009; Schultz, 2011). Without the elimination of chronic stimulants such as tobacco, permanent recovery from dysphonia is difficult to attain through surgical operations alone (Boone et al., 2013; Cohen, 2010). Therefore, for the prevention of dysphonia, risk factors must be identified and systematically managed.

Meanwhile, self-reported voice problems have been identified as a significant risk indicator for the progression of dysphonia (Byeon, 2011; Miller & Verdolini, 1995). Together with the laryngeal endoscopy test and acoustic phonetic test, the self-reported status of a vocal problem is a standard in the diagnosis of dysphonia in clinical practice (Lehto et al., 2006; Verdolini & Ramig, 2001). Although it has a tendency to be perceived as a more severe problem than it actually is, the subjective recognition of health problems has long been used as an indicator to represent the health status of interviewees because it is easy to measure and useful for predicting the risk of disease (Idler & Kasl, 1991; Ware, 1986).

Even though several studies (Feierabend & Shahram, 2009; Marcotullio, Magliulo & Pezone, 2002; Zvrko, Gledović & Ljaljević, 2008) have reported that smoking has an adverse effect on the larynx, two studies on community populations did not find any significant association between smoking and dysphonia (Byeon, 2014; Roy et al., 2005). One of the studies (Byeon, 2014) diagnosed laryngeal disorders with only the diagnostic results of a laryngeal endoscopy; thus, it cannot be concluded that smoking does not have anything to do with voice problems simply because it was confirmed that smoking has no association with laryngeal disorders. As it currently stands, there have been few, if any, studies on the relationship between the amount people have smoked over their lifetime and voice problems in the community population.

This study aims to investigate the relationship between current smoking and lifetime amount smoked and the prevalence of dysphonia using data from a national cross-sectional survey that represents the Korean population.

Methods

Subjects

The data resource of this study was part of raw data of The Korea National Health and Nutrition Examination Survey (KNHANES) conducted by Korea Centers for Disease Control and Prevention in 2008. The KNHANES is a national cross-sectional survey for the non-institutionalized civilian population of South Korea. A complex, stratified, multistage probability sampling design based on age, sex, and region was used in this survey to represent the Korean population (Ministry of Health and Welfare, 2010). The survey was approved by the Institutional Review Board of the Korean Center for Disease Control and Prevention (2011-02CON-06). In brief, The KNHANES consists of health questionnaire survey, examination survey and nutrition survey. The health questionnaire survey included health behaviors such as smoking and drinking; for the Examination Survey, otolaryngological examinations were conducted by otolaryngologists. Surveys on level of education and economic activities were conducted with individual interviews, while health behaviors like smoking and self-reported voice problems were surveyed by self-administered reporting method. The target population of KNHANES comprises non-institutionalized Korean citizens residing in Korea. The sampling plan follows a multi-stage clustered probability design based on the National Census Data. For example, in the 2008 survey, 100 primary sampling units were drawn from approximately 200,000 geographically defined primary sampling units for the whole country. The 2008 KNHANES response rate was 74.3% for the health interview and examination survey.

The subjects of this study were 3,623 adults over the age of 19 who completed both the Health Questionnaire Survey and an otolaryngological examination of KNHANES performed in 2008. Among them, finally a total 3,600 adults (1,501 males and 2,099 females) were analyzed with the exclusion of 32 non-respondents to otolaryngologic interviews (Fig. 1).

Figure 1 The flow chart of the study.

Measurement

Smoking

Smokers were classified into current smokers, past smokers and non-smokers. According to the definition of WHO, a current smoker was defined as someone who has smoked more than 100 cigarettes over their life and either sometimes smokes, or smokes every day. A past smoker is defined as someone who has smoked more than 100 cigarettes during their life and does not currently smoke. Pack years (total smoked over a lifetime) was calculated as packs of cigarettes smoked per day multiplied by the years spent as a smoker (Sabia et al., 2008). Smokers were grouped based on their calculated pack years, into light smoking (≤26.7 pack years), medium smoking (26.7–40.5 pack years), heavy smoking (40.5–55.5 pack years), and very heavy smoking (55.5–156 pack years) with reference to preceding studies (Juan et al., 2004; Tyas et al., 2003).

Self-reported voice problems

Self-reported voice problems were classified into ‘yes’ and ‘no’ in response to the question ‘Do you think that you currently have pain and discomfort in your own voice?’ in the otolaryngologic interviews.

Laryngeal disease

A total of 45 otolaryngologists from 43 general hospitals conducted endoscopic laryngeal examinations for laryngeal lesions using a 70° endoscope that was attached to a charge-coupled device camera on male and female adults aged 19 and over.

The laryngeal examinations were in collaboration with the Korean Society of Otorhinolaryngology-Head and Neck Surgery, who provided technical advice and highly trained otolaryngologists. Prior to the research, otolaryngologists were trained three times, and common criterion errors were examined through theory education, pre-training, and mock surveys. The index of coincidence was evaluated twice, and the quality improvement committee re-evaluated the examined pictures and videos (640 × 480-sized audio video interleave files which were compressed by DivX 4.12 codec using a compression rate of 6 Mb/s) by otolaryngologists and computed results. The laryngoscope examination index of coincidence was 75%.

Based on data from laryngoscopic examination, the definition of laryngeal disease included vocal nodules, vocal polyp, intracordal cyst, Reinke edema, laryngeal granuloma, sulcus vocalis, laryngeal keratosis, laryngitis, laryngeal papilloma, vocal cord paralysis, and suspected malignant neoplasm of the larynx.

Confounding variables

Confounding variables included age, gender, level of education, income, occupation, alcohol drinking, pain & discomfort from disease for the recent 2 weeks. Level of education was classified into primary, middle and high school graduates and university graduates and over. Income was classified into 4 quartiles. Occupation was surveyed based on the 6th revised version of the Korean Standard for Classification of Occupations (KSCO-6; Korea National Statistical Office, 2007) and was reclassified into economically inactive (unemployed persons, housewives and students), non-manual (managers & professionals, clerical support workers and service & sales workers) and manual (skilled agricultural & forestry & fishery workers, craft & related trades workers and elementary occupations). Alcohol drinking was classified into ‘less than once a month’ and ‘once or more’ a month based on the drinking behavior for recent one year. Pain and discomfort from disease for the recent 2 weeks was classified into ‘yes’ and ‘no.’

Statistical analysis

The weighted values of The KNHANES were calculated so that the subjects of the survey could represent the overall Korean population. Detailed explanations of how the weighted values were derived are shown in Ministry of Health and Welfare (2010). For the characteristics of subjects based on whether or not they smoked and the total amount smoked over their lifetime, the weighted mean, standard variance and weighted percentage were presented using descriptive analysis while, for the difference between groups, a continuous variable was analyzed with a weighted one-way ANOVA and the nominal variable was analyzed with a Rao-Scott chi-square test. For the association between pack years and dysphonia, the odds ratio and 95% confidence interval were presented using hierarchical logistic regression. For confounding variables, only socio-demographic variables were adjusted for Model 1; both socio-demographic variables and health-risk behavior variables were adjusted for Model 2, and all confounding variables including level of health were adjusted for Model 3. IBM SPSS version 21.0 (IBM Inc., Chicago, Illinois, USA) was used for all analyses and significance level was 0.05 in two-sided test.

Results

Characteristics of subjects based on level of pack years

The characteristics of subjects based on the level of pack years are demonstrated in Table 1. The average age was the lowest for light smokers and highest for very heavy smokers (p < 0.001). Males had the highest percentage of heavy smokers and the lowest percentage of non-smokers, whereas females had the highest percentage of non-smokers and the lowest percentage of heavy smokers (p < 0.001). As for level of education, a high percentage of heavy smokers and very heavy smokers were elementary school graduates, while a high percentage of light smokers were high school and university graduates (p < 0.001). As for occupation, non-smokers included the highest percentage of economically inactive individuals, while medium and heavy smokers included the highest percentage of manual laborers (p < 0.001). Concerning drinking, non-smokers included the highest percentage of those who drank less than once a month, whereas light, medium, and heavy smokers included a higher percentage of those who drank once or more a month (p < 0.001). Regarding level of health, very heavy smokers had the highest level of recognition of pain and discomfort from disease over the last two weeks, self-reported voice problems and laryngeal disease (p < 0.05).

Table 1 Characteristics of subjects based on level of packyears, weighted %.

Characteristics	Non smoke	Light	Medium	Heavy	Very heavy	p *	
	(n = 2, 114)	(n = 1, 372)	(n = 142)	(n = 41)	(n = 17)		
Age (mean ± s.d)	49.1 ± 16.6	47.3 ± 16.6	54.4 ± 13.6	56.9 ± 12.3	63.6 ± 12.2	<0.001	
Sex						<0.001	
Male	13.8	82.2	89.4	90.2	76.5		
Female	86.2	17.8	10.6	9.8	23.5		
Education level						<0.001	
Elementary school	34.1	21.3	33.8	46.3	52.9		
Middle school	10.3	11.6	16.9	19.5	17.6		
High school	32.9	38.2	31.7	22.0	11.8		
≥ College	22.6	28.9	17.6	12.2	17.6		
Income						0.744	
1st quartile	21.8	18.5	19.3	22.5	31.2		
2nd quartile	26.5	27.2	27.9	25.0	25.0		
3rd quartile	27.0	28.8	28.6	22.5	31.2		
4th quartile	24.6	25.5	24.3	30.0	12.5		
Occupation						<0.001	
Economically inactive	48.0	28.6	27.5	27.5	52.9		
Non-manual	26.0	36.3	26.1	20.0	17.6		
Manual	26.0	35.1	46.5	52.5	29.4		
Alcohol drinking						<0.001	
>1 time per month	60.7	24.8	27.5	26.8	52.9		
≥1 time per month	39.3	75.2	72.5	73.2	47.1		
Pain and discomfort during the last 2 weeks						<0.001	
Yes	31.7	21.6	20.4	34.1	58.8		
Laryngeal disease						<0.001	
Yes	5.3	7.5	9.2	9.8	23.5		
Self-reported voice problems						0.010	
Yes	6.8	6.2	3.5	17.1	17.6		
Notes.

* Rao-Scott chi-square test for categorical variables; weighted ANOVA test for continuous variables.

The light smoking defined as ≤26.7 pack years; medium smoking defined as >26.7–40.5 pack years; heavy smoking defined as >40.5–55.5 pack years; very heavy smoking defined as >55.5–156 pack years.

Association between current smoking and dysphonia

The association between current smoking habits and dysphonia is presented in Table 2. The results of the hierarchical logistic regression, where socio-demographic variables (age, gender, level of education and occupation) were adjusted in Model 1, indicated that current smokers had a 1.02 times (OR = 1.02, 95% CI [1.01–1.03]) higher risk of developing voice problems than non-smokers. Moreover, current smokers had a 1.5 times (OR = 1.53, 95% CI [1.01–2.33]) higher risk of laryngeal disorder (p < 0.05).

Table 2 Hierarchical logistic regression analyses of the association between smoking and dysphonia: odds ratio (OR) and confidence interval (CI).

Smoking	Model 1	Model 2	Model 3	
	SVP	LD	SVP	LD	SVP	LD	
Non smoker	1	1	1	1	1	1	
Past smoker	1.37	1.05	1.42	1.06	1.38	1.06	
	(0.89, 2.12)	(0.66, 1.65)	(0.92, 2.19)	(0.69, 1.67)	(0.90, 2.14)	(0.67, 1.68)	
Current smoker	1.02	1.53	1.81	1.56	1.77	1.56	
	(1.01, 1.03)*	(1.01, 2.33)*	(1.19, 2.73)*	(1.02, 2.38)*	(1.17, 2.68)*	(1.02, 2.38)*	
Notes.

* p < 0.05

SVP self-reported voice problem

LD laryngeal disease

Model 1 adjusted for age, sex, education, quartiles of income, and occupation

Model 2 additionally adjusted for alcohol drinking

Model 3 additionally adjusted for pain and discomfort during the last 2 weeks

When drinking was additionally adjusted in Model 2, current smokers had a 1.8 times (OR = 1.81, 95% CI [1.19–2.73]) higher risk of developing self-reported voice problems than non-smokers, and current smokers had a 1.6 times (OR = 1.56, 95% CI [1.02–2.38]) higher risk of laryngeal disorder (p<0.05).

In Model 3, when all confounding variables were adjusted, current smokers had a 1.8 times (OR = 1.77, 95% CI [1.17–2.68]) higher risk of self-reported voice problems than non-smokers, and current smokers had a 1.6 times (OR = 1.56, 95% CI [1.02–2.38]) higher risk of laryngeal disorder (p<0.05).

Association between pack years and dysphonia

The association between pack years and dysphonia for smokers is provided in Table 3. As a result of the hierarchical logistic regression, heavy smoking (40.5–55.5 pack years) was found to have a significant relationship with self-reported voice problems, and very heavy smoking (>55.5–156) was found to have a significant relationship with laryngeal disorder.

Table 3 Hierarchical logistic regression analyses of the association between the Level of smoking and dysphonia: odds ratio (OR) and confidence interval (CI).

Level of smoking (pack years)	Model 1	Model 2	Model 3	
	SVP	LD	SVP	LD	SVP	LD	
Light	1.38	1.18	1.47	1.20	1.44	1.20	
(≤26.7)	(0.95, 2.01)	(0.81, 1.71)	(1.00, 2.14)	(0.82, 1.76)	(0.98, 2.10)	(0.82, 1.76)	
Medium	0.77	1.34	0.82	1.37	0.84	1.36	
(>26.7–40.5)	(0.30, 1.95)	(0.70, 2.55)	(0.32, 2.09)	(0.71, 2.62)	(0.33, 2.15)	(0.71, 2.61)	
Heavy	3.69*	2.13	3.96*	2.18	3.86*	2.17	
(>40.5–55.5)	(1.64, 8.31)	(0.83, 5.44)	(1.75, 8.95)	(0.85, 5.58)	(1.69, 8.79)	(0.85, 5.57)	
Very heavy	3.05	3.98*	3.19	4.01*	2.70	3.98*	
(>55.5–156)	(0.95, 9.83)	(1.22, 12.95)	(0.99, 10.28)	(1.23, 13.05)	(0.84, 8.73)	(1.22, 13.00)	
Notes.

Reference group is non smoker

* p < 0.05, pack years = (packs smoked per day) × (years as a smoker).

SVP self-reported voice problem

LD laryngeal disease

Model 1 adjusted for age, sex, education, quartiles of income, and occupation.

Model 2 additionally adjusted for alcohol drinking

Model 3 additionally adjusted for pain and discomfort during the last 2 weeks.

When all confounding variables were adjusted (Model 3), heavy smokers had a 3.9 times (OR = 3.86, 95% CI [1.69–8.79]) higher risk of self-reported voice problems than non-smokers, and very heavy smokers had a 4 times (OR = 3.98, 95% CI [1.22–13.00]) higher risk of laryngeal disorder (p < 0.05).

Discussion

This national cross-sectional study analyzed the relationship between current smoking and pack years and dysphonia for adults in local communities. The study demonstrated that current smokers had a 1.8 times (OR = 1.77, 95% CI [1.17–2.68]) higher risk of self-reported voice problems than non-smokers, and current smokers had a 1.6 times (OR = 1.56, 95% CI [1.02–2.38]) higher risk of laryngeal disorder. This result was incongruous with that of Glas et al. (2008) who concluded that smoking did not have a significant effect on self-reported voice problems. This difference may be due to the fact that the study of Glas et al. (2008) had as its subjects only patients with vocal disorders who visited medical institutions. Naturally, no significant relationship between self-reported voice problems and smoking was shown, since most patients already had voice problems when they visited medical institutions whether they smoked or not.

The causal relationship between smoking and vocal disorders has been corroborated in diverse studies. Numerous epidemiological studies have reported that smoking is a major cause of Reinke’s edema (Marcotullio, Magliulo & Pezone, 2002; Zvrko, Gledović & Ljaljević, 2008) and is closely linked with chronic laryngitis, laryngeal keratosis and laryngeal leukoplakia (Feierabend & Shahram, 2009). Smoking has been proven to be the direct cause of deformation in the mucosa of vocal cords in animal experimental research studies as well (Duarte et al., 2006; Işik et al., 2004). Chronic smoking causes anatomical changes in the larynx, worsening pain and voice problems (Park, 2003). Furthermore, since smoking causes vocal fatigue in the short term by stimulating the mucosal membrane of the vocal cords, it is speculated that current smokers are more likely to perceive their own voice problems than non-smokers. Since few studies have analysed the relationship between smoking and self-reported voice problems in local community populations, cohort studies on various races should be conducted in the future.

Another finding of this study is that in terms of pack years, very heavy smokers were significantly more likely to suffer from laryngeal disorder, while self-reported voice problems were significantly more likely in heavy smokers but not very heavy smokers. Preceding studies (Lubin et al., 2007; Lubin et al., 2008) have reported that an increase in pack years subsequently increases the risk of laryngeal pathologies such as larynx cancer. However, even those classed as very heavy smokers in terms of pack years in this study were not significantly more likely to suffer from self-reported voice problems. This can be explained by two factors: First, it is possible that smokers’ subjective recognition of voice problems did not reflect their actual level of health. Even though subjective recognition of voice problems is a useful index that represents a subject’s overall health status, subjects tend to over- or under-estimate the actual status of their health (Idler & Kasl, 1991). Thus, it is possible that no significant relationship was found between very heavy smoking and self-reported voice problems since very heavy smokers with excessive pack years perceived their vocal health as better than it actually was. Second, the result may have been due to the survival effect of very heavy smokers. The average age of very heavy smokers was over 60, which was the highest of all groups. The subjects of this study were from the general population of local communities, and thus those with poor health from heavy smoking might have been hospitalized or already deceased.

The limitations of this study are as follows: First, as this study investigated only the subjective recognition of voice problems, it is not possible to interpret the severity and type of voice problems. Thus, future studies that use standardized subjective vocal evaluation methods are required. Second, since smoking is a negative health behavior, survey responses might have been underestimates. Third, since this is a cross-sectional study, the results cannot be interpreted as indicating a causal relationship.

Conclusion

Excessive cigarette smoking had an independent relationship with dysphonia. Further longitudinal studies are required to investigate the causality between smoking and voice problems.

The author would like to thank the Korea Centers for Disease Control and Prevention for providing data and consultations.

Additional Information and Declarations

Competing Interests

Author Contributions

Human Ethics

The author declares there are no competing interests.

Haewon Byeon conceived and designed the experiments, performed the experiments, analyzed the data, contributed reagents/materials/analysis tools, wrote the paper, prepared figures and/or tables, reviewed drafts of the paper.

The following information was supplied relating to ethical approvals (i.e., approving body and any reference numbers):

The survey was approved by the Institutional Review Board of the Korean Center for Disease Control and Prevention (2011-02CON-06).

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
