# Peer review of "The association between lifetime cigarette smoking and dysphonia in the Korean general population: findings from a national survey"

_PeerJ, doi:10.7717/peerj.912_

## Round 0.1 · original submission · Major Revisions

The authors in this study analysed the association between lifetime cigarettes smoking and self-reported voice problems in Korean general population. After careful consideration, this submission cannot be accepted in this current form. The authors have to improve the content and the concepts of this manuscript by performing the comments highlighted by the Reviewers.

Reviewer 1 ·

Basic reporting

Byeon in this paper investigate the relationship between current smoking & lifetime amount smoked and the incidence of self-reported voice problems using national cross-sectional survey that can represent the Korean population.

Comments:
- I suggest the Authors to use an English more appropriate.
- In the Material and Methods section there isn’t in my opinion an accurate description about the sample analyzed. Which were the inclusion and the exclusion criteria used in the choice of the sample analyzed?

Experimental design

"No Comments".

Validity of the findings

"No Comments".

Additional comments

"No Comments".

Reviewer 2 ·

Basic reporting

Byeon H in this manuscript analysed the association between lifetime cigarettes smoking and self-reported voice problems in Korean general population. At this aim data resourece for this study was part of the raw data of the Korea national health and Nutrition Examination Survey (KHANES) conducted by Korea center for Disease Control and Prevention in 2008.
In the Methods section the author reported that the KHANES consists of health questionnaire survey, examination survey and nutrition survey. Furthermore, he reported that the Health questionnaire survey included health behaviors such as smoking and drinking and for Examination Survey, otolaryngological examinations by otolaryngologists, survey on level of education and economic activities with individual interviews, while health behaviors like smoking and self-reported voice problems were surveyed by self-administered reporting method.
Subjects included in the study were 3.600 adults.

Experimental design

In the methods section the author reported measures performed and among these it is possible to find the following paragraphs: Smoking; Self-reported voice problems; Confounding factors

In the results section the author reported the following paragraphs are reported:

- Characteristics of subjects' self-reported voice problems based on the level of pack years
- Association between current smoking and self-reported voice problems
- Association between pack years and self-reported voice problems

Validity of the findings

The question is why the author did not report any data about otolaryngological examinations by otolaryngologists? This is an important issue because the same author in the discussion section say that “this study investigated only the subjective recognition of voice problems, it is not possible to interpret the severity and type of voice problems, for which future studies are required that use standardized subjective vocal evaluation methods.” and put it as a limit of the study. I think that data about otolaryngological examinations by otolaryngologists should bbe useful to overcome this limit. Furthermore, the author affirms that “there is the possibility that smokers' subjective recognition of voice problems did not reflect the actual level of their health”. To have a more objective evaluation of data reported it is important to evaluate this issue.

Additional comments

In general I think that the study should be interesting but information requested about otolaryngological examinations by otolaryngologists are needed and these data have to be evaluated together wioth other factors such as number of cigarettes packs, economic and cultural status. Please, provide these data.

---

## Round 0.2 · accepted · Accept

Dear Dr Haewon Byeon,

Thank you for submitting your manuscript to PeerJ. After careful consideration, we have decided to accept your manuscript. You performed the revision required and you give all information required.

Only a minor comment, in the result section in the section “Association between current smoking and dysphonia” it is better to replace the sentence “were adjusted in Model 1, indicated that current smokers had a 102% (OR=1.02, 95% CI: 1.01–1.03) higher risk of developing voice problems than non-smokers.”

with the sentence:

“were adjusted in Model 1, indicated that current smokers had a 1.02 times (OR=1.02, 95% CI: 1.01–1.03) higher risk of developing voice problems than non-smokers.”

Best regards.

Dr. Alessandra Lo Presti

Reviewer 2 ·

Basic reporting

.

Experimental design

.

Validity of the findings

.

Additional comments

The topic of this manuscript falls within the scope of Peer Journal.

Byeon H in this manuscript analysed the association between lifetime cigarettes smoking and self-reported voice problems in Korean general population. At this aim data resourece for this study was part of the raw data of the Korea national health and Nutrition Examination Survey (KHANES) conducted by Korea center for Disease Control and Prevention in 2008.
Authors revised their manuscript following the suggestion given in the revision.